# SAND: One-Shot Feature Selection with Additive Noise Distortion

**Pedram Pad** [1]  **Hadi Hammoud** [1 2]  **Mohamad Dia** [1]  **Nadim Maamari** [1]  **L. Andrea Dunbar** [1 2]

## Abstract

Feature selection is a critical step in data-driven applications, reducing input dimensionality to enhance learning accuracy, computational efficiency, and interpretability. Existing state-of-the-art methods often require post-selection retraining and extensive hyperparameter tuning, complicating their adoption. We introduce a novel, non-intrusive feature selection layer that, given a target feature count $k$, automatically identifies and selects the $k$ most informative features during neural network training. Our method is uniquely simple, requiring no alterations to the loss function, network architecture, or post-selection retraining. The layer is mathematically elegant and can be fully described by:

$$\tilde{x}_i = a_i x_i + (1 - a_i) z_i$$

where $x_i$ is the input feature, $\tilde{x}_i$ the output, $z_i$ a Gaussian noise, and $a_i$ trainable gain such that $\sum_i a_i^2 = k$. This formulation induces an automatic clustering effect, driving $k$ of the $a_i$ gains to 1 (selecting informative features) and the rest to 0 (discarding redundant ones) via weighted noise distortion and gain normalization. Despite its extreme simplicity, our method achieves competitive performance on standard benchmark datasets and a novel real-world dataset, often matching or exceeding existing approaches without requiring hyperparameter search for $k$ or retraining. Theoretical analysis in the context of linear regression further validates its efficacy. Our work demonstrates that simplicity and performance are not mutually exclusive, offering a powerful yet straightforward tool for feature selection in machine learning.

[1]CSEM, Neuchâtel, Switzerland [2]EPFL, Lausanne, Switzerland. Correspondence to: Pedram Pad <pedram.pad@csem.ch>.

*Proceedings of the 42$^{nd}$ International Conference on Machine Learning*, Vancouver, Canada. PMLR 267, 2025. Copyright 2025 by the author(s).

## 1. Introduction

Feature selection is a fundamental problem in high-dimensional statistics and machine learning (Guyon & Elisseeff, 2003; Li et al., 2017). Unlike feature extraction techniques that alter features' semantics by creating new ones in a lower dimensional space, feature selection involves the identification and retention of the most informative features while discarding irrelevant or redundant ones. This preservation enhances the interpretability and explainability of predictive models, particularly critical in domains like medicine and biology where gene selection is a focal application (Guyon et al., 2002). By retaining the original features, researchers can directly relate model outputs to the underlying data, facilitating insights and hypothesis generation. Furthermore, feature selection not only contributes to storage reduction by eliminating unnecessary data points, optimizing memory usage, and enhancing computational efficiency, but also aids in reducing model size and complexity. By selecting a subset of input features, models can improve performance and generalization capabilities crucial for mitigating overfitting and addressing the curse of dimensionality. Moreover, in applications where sensing hardware costs or energy consumption are major concerns, such as in IoT devices or sensor-based systems, feature selection can inform the design of simpler and more cost-effective hardware by ensuring that only relevant features are sensed or measured, thereby conserving resources without compromising performance.

Feature selection methods can be broadly categorized into two main groups: unsupervised and supervised. Unsupervised methods often involve an analysis of the relations between input features through methods like clustering (He et al., 2005), matrix factorization (Wang et al., 2015), and the use of autoencoder neural networks (Balın et al., 2019). These methods are useful when labeled data is scarce or unavailable, allowing for the exploration of inherent data structures and patterns. On the other hand, supervised methods leverage the availability of labeled data to guide the selection process. Within the realm of these methods, there exist model-independent and model-dependent approaches. Model-independent methods, also known as filter-based, rely on statistical tests and information-theoretic metrics to evaluate feature relevance with respect to the target variable, irrespective of the underlying machine learning model

(Yang & Pedersen, 1997; Chandrashekar & Sahin, 2014). While these methods are computationally efficient and can handle high-dimensional data, they may overlook complex interactions between features. Model-dependent methods, on the other hand, tailor feature selection to specific machine learning models or architectures. This category can be further divided into wrapper and embedded methods. Wrapper methods (Kohavi & John, 1997) involve a search process guided by the final performance of a learning model, such as classifier accuracy. Examples include greedy sequential feature selection (Das & Kempe, 2011), SHAP (SHapley Additive exPlanations) values calculation (Lundberg & Lee, 2017), in addition to combinatorial optimization and meta-heuristic search algorithms (Zadeh et al., 2017; Dokeroglu et al., 2022). Wrapper methods offer the advantage of considering feature interactions but may suffer from high computational costs due to intensive search over the input space, which is highly impractical for complex models and large feature dimensions. In contrast, embedded methods rank features based on metrics intrinsically learned during model training, seamlessly integrating feature selection into the learning process. Examples include feature importance for tree-based algorithms (Breiman, 2001), Recursive Feature Elimination for Support Vector Machine (RFE-SVM) (Guyon et al., 2002), sparsity-promoting models (Tibshirani, 1996), and deep learning techniques (Simonyan et al., 2014; Wang et al., 2014). Such methods enable an automatic selection of relevant features during training and can effectively handle non-trivial relationships in data.

### Related works

Given the pervasive adoption of deep learning in recent years, this work concentrates on embedded feature selection techniques tailored to neural networks. Within this domain, a multitude of approaches have emerged, predominantly centered around various adaptations of LASSO-based regularization (Luo & Chen, 2014; Zhao et al., 2015; Li et al., 2016; Lemhadri et al., 2021; Cancela et al., 2023), the addition of stochastic gates (Srinivas et al., 2017; Borisov et al., 2019; Yamada et al., 2020), the use of attention mechanisms (Liao et al., 2021; Yasuda et al., 2023), and the application of saliency maps (Cancela et al., 2020) to solve the feature selection problem on non-linear models. For instance, Sequential LASSO (Luo & Chen, 2014) provides an efficient implementation of greedy LASSO to recursively select input features, while Group LASSO (Zhao et al., 2015; Scardapane et al., 2017) further modifies the objective function to encourage sparsity at the group level. In LassoNet (Lemhadri et al., 2021), a skip linear connection is added to the neural network with two types of regularization parameters. A continuous search is then applied using a hierarchical proximity algorithm, which combines a proximal gradient descent method with a hierarchical feature

selection. Alternatively, to impose sparsity and overcome the limitations of applying gradient descent on $\ell_1$ regularized objective functions, (Yamada et al., 2020) introduces a continuous relaxation of Bernoulli gates that are attached to the input features. A Gaussian-based regularization is then added to the objective function and grid-search over the regularization parameter is applied to select the required number of features. While (Yamada et al., 2020) employs a similar stochastic approach as ours, it modifies the loss function by adding an additional term and lacks direct control over the number of selected features, requiring a grid search over a model hyperparameter. In contrast, our stochastic framework does not alter or rely on the loss function, eliminating the need for retraining or hyperparameter tuning. Additionally, our method allows direct specification of the desired number of features as an input. Lately, the attention mechanism is being employed to relate a trainable softmax mask to feature importance, and hence perform embedded feature selection by adaptively estimating marginal feature gains over multiple rounds (Yasuda et al., 2023).

### Contributions

The existing methods mentioned above typically necessitate alterations to the objective function or significant modifications to the neural network architecture involving the addition of new connections. Consequently, feature selection is often a separate phase followed by a retraining phase on the selected features, or it requires some kind of hyperparameter tuning to control the number of selected features (Yamada et al., 2020; Lemhadri et al., 2021; Yasuda et al., 2023). In this work, we propose a novel, yet exceptionally simple, method for one-shot feature selection. It involves the integration of a simple constrained weighted additive noise layer at the neural network's input. The constrained stochasticity helps the network generate a polarized input space and effectively select the desired number of features during training. As a result, the network architecture inherently converges to its final form, which can be used for inference without necessitating any additional retraining. The constraint on the weights is imposed by construction through a normalization operation and requires no regularization terms in the objective function. The proposed layer imposes negligible computational overhead and can be seamlessly incorporated, akin to the addition of Dropout or Batch Normalization layers. Through this layer, direct control over the number of selected features is enabled without the need for additional grid search or further tuning of regularization terms. The simplicity of our method does not compromise the final prediction performance of the neural network. In this work, we conduct an extensive benchmarking study against state-of-the-art feature selection methods using common datasets and a novel real-world dataset, showcasing our method's effective competition against existing approaches

while highlighting its practicality and application-driven nature. Furthermore, we provide theoretical insights by demonstrating that our method, when applied to linear regression, promotes the selection of a predefined number of features on an equivalent problem.

## 2. Selection with Additive Noise Distortion (SAND)

In a typical supervised learning problem, we are tasked to map a set of input vectors to predefined outputs. These input vectors consist of various features. However, not all features are equally important in determining the output. Some may be irrelevant, while others might contain redundant information. This leads us to the concept of feature selection: the quest to identify the subset of features that provide sufficient information to determine the output accurately. In real-world applications, the number of features to select is typically pre-defined due to constraints on data acquisition burden, computation cost, or memory footprint.

Consider an $n$-dimensional feature vector $\underline{x} = (\boldsymbol{x}_1, \boldsymbol{x}_2, \ldots, \boldsymbol{x}_n)^\top$ (which can be of any shape; but for the sake of simplicity in notations, we assume it to be $n \times 1$) to be mapped to the output vector $\underline{y}$[1]. Figure 1(a) depicts a typical neural network solution to this problem. Now, assume we are interested in finding the $k$ dimensions that yield the highest performance, with $k \leq n$.

Our idea is to multiply each feature $\boldsymbol{x}_i$ with a gain $a_i$ and add a zero-mean Gaussian noise with the standard deviation of $|(1 - a_i)\sigma|$ to it before feeding it to the neural network. Here, $\sigma$ is a fixed scalar. Moreover, we constrain the vector $\underline{a} = (a_1, a_2, \ldots, a_n)^\top$ to have the $\ell_\alpha$-norm equal to $k^{\frac{1}{\alpha}}$ for a pre-selected $\alpha > 0$. Thus, we define

$$\tilde{\underline{x}} = \underline{a} \odot \underline{x} + (\underline{1} - \underline{a}) \odot \underline{z} \tag{1}$$

where

$$\|\underline{a}\|_\alpha^\alpha = k. \tag{2}$$

We then feed $\tilde{\underline{x}}$ to the neural network during the training phase instead of $\underline{x}$ as illustrated in Figure 1(b)). Here, $\underline{z}$ is a Gaussian vector with i.i.d. entries with zero-mean and standard deviation $\sigma$. In this setting, when $a_i$ is close to 1, $\tilde{x}_i$ is close to noiseless $\boldsymbol{x}_i$, and when $a_i$ is close to 0, $\tilde{x}_i$ becomes almost pure noise (the signal-to-noise ratio is proportional to $\frac{a_i^2}{(1-a_i)^2\sigma^2}$). During the training phase, we allow the $a_i$'s to be trained alongside the other parameters of the network. The architecture of the neural network and the loss function remain unchanged; the only difference is the

---

[1]Throughout the paper, we use small characters to denote scalars, underlined characters to indicate vectors, capital characters for matrices and bold font for random objects

addition of $n$ extra parameters ($a_i$'s) to optimize. As training progresses, we observe that $k$ of the $a_i$'s cluster around 1, indicating the selected features, while the remaining $a_i$'s cluster around 0, indicating the neglected features. We refer to this approach as SAND, which stands for Selection with Additive Noise Distortion.

*Remark* 2.1. The two operations of the SAND layer in (1) and (2) are implemented together. This means the constraint (2) is enforced by construction where we normalize the $a_i$'s by their $\ell_\alpha$-norm inside the layer, without adding any regularization term to the loss function. Hence, the SAND layer takes this form in practice:

$$\tilde{\underline{x}} = \frac{\underline{a}}{\|\underline{a}\|_\alpha} k^{\frac{1}{\alpha}} \odot \underline{x} + \left(\underline{1} - \frac{\underline{a}}{\|\underline{a}\|_\alpha} k^{\frac{1}{\alpha}}\right) \odot \underline{z}. \tag{3}$$

Notice that if there is no noise $\underline{z}$ (i.e., $\sigma = 0$), the $a_i$'s would be absorbed in the weights of the first layer of the neural network. Additionally, if $k = n$, all $a_i$'s can become 1 and then $\tilde{\underline{x}}$ will be identical to $\underline{x}$ without any noise. Given that the noise is independent of the data and lacks information about the output, the network naturally adjusts to mitigate its impact during training.

The first non-trivial property is that there is always an optimal $\underline{a}$ whose entries are between 0 and 1. Here, optimal means with respect to the loss function of the network. To prove that, assume there is an optimal $\underline{a}$ that has an $a_j < 0$. By replacing $a_j$ by $-a_j$, while the constraint (2) still holds, $\tilde{x}_j$ is a less noisy version of $\boldsymbol{x}_j$. On the other hand, if there is an optimal $\underline{a}$ that has an $a_j > 1$, we can decrease $a_j$ to 1, which results in $\tilde{x}_j$ becoming a noiseless copy of $\boldsymbol{x}_j$, and increase other $a_i$'s that are less than 1 towards 1 to satisfy the condition (2); it decreases the noise added to those features as well. Therefore, we can confine the search space of $\underline{a}$ to the vectors that have all entries between 0 and 1, i.e., $\underline{a} = (a_1, a_2, \ldots, a_n)^\top$ that have

$$0 \leq a_i \leq 1 \quad \text{for } i = 1, \ldots, n. \tag{4}$$

Now, we analyze what is happening during the training. Intuitively, a more informative feature $x_j$ will get a gain $a_j$ closer to 1 so that it will be passed to the neural network with less noise. Due to the constraint (2), this automatically yields smaller gains $a_{j'}$ for other features which are less informative. Consequently, the less informative features become noisier, which makes them even less informative and pushes them to get even smaller gains. This leaves more room, due to (2), for the more informative features to get their gains closer to 1 and become less noisy. This reinforcing loop, summarized in Figure 2, results in polarization of the gains around 1 and 0. Ideally, we will end up with a vector $\underline{a}$ which has $k$ entries equal to 1, indicating the selected features, and the rest equal to 0, indicating the neglected features. However, since in practice the smallest

SAND Layer

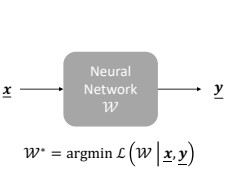

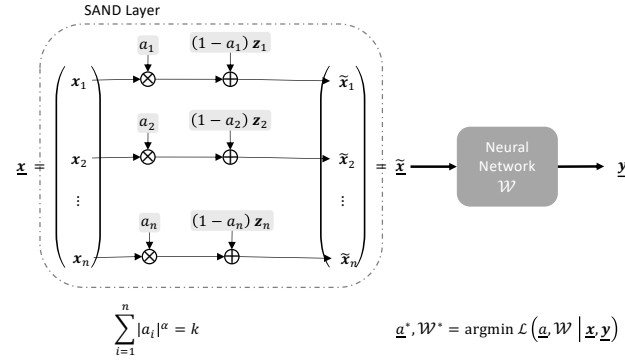

(a) Vanilla Neural Network

(b) Neural Network with the SAND layer

*Figure 1.* Neural network architecture and the loss function before and after adding the SAND layer. Here, $\mathcal{L}$ and $\mathcal{W}$ indicate the loss function and the trainable parameters of the neural network respectively.

values have not necessarily converged to absolute zero, at the end of the training phase, we keep the top $k$ gains intact and manually set the $n - k$ smallest gains to 0, effectively removing those features. Notably, features with small gains are noisy and hence inherently ignored by other parts of the neural network.

**Linear Regression**

Here, we mathematically show that in the case of linear regression, adding the SAND layer introduced above is equivalent to adding a term in the loss function that promotes selection of $k$ features.

In linear regression problem, the loss function is

$$\mathcal{L}(W, \underline{b}) = \mathbb{E}_{\underline{x}, \underline{y}} \left\{ \left\| \underline{y} - W\underline{x} - \underline{b} \right\|_2^2 \right\}, \qquad (5)$$

where $\mathbb{E}$ denotes the expected value, W is the coefficient matrix and $\underline{b}$ is the bias vector. Thus, optimal solution is

$$W^*, \underline{b}^* = \underset{W, \underline{b}}{\operatorname{argmin}} \, \mathcal{L}(W, \underline{b}). \qquad (6)$$

Now, we add the SAND layer in the beginning, i.e.,

$$\underline{\tilde{x}} = \underline{a} \odot \underline{x} + (\underline{1} - \underline{a}) \odot \underline{z} \quad \text{such that} \quad \|\underline{a}\|_\alpha^\alpha = k. \quad (7)$$

We get

$$\mathcal{L}(\underline{a}, W, \underline{b}) \qquad (8)$$

$$= \mathbb{E}_{\underline{\tilde{x}}, \underline{y}} \left\{ \left\| \underline{y} - W\underline{\tilde{x}} - \underline{b} \right\|_2^2 \right\}$$

$$= \mathbb{E}_{\underline{x}, \underline{y}, \underline{z}} \left\{ \left\| \underline{y} - W(\underline{a} \odot \underline{x} + (\underline{1} - \underline{a}) \odot \underline{z}) - \underline{b} \right\|_2^2 \right\}$$

$$= \mathbb{E}_{\underline{x}, \underline{y}} \left\{ \left\| \underline{y} - W(\underline{a} \odot \underline{x}) - \underline{b} \right\|_2^2 \right\} + \sum_{i=1}^n w_i^2 (1 - a_i)^2 \sigma^2$$

where $w_i$ is the $\ell_2$-norm of the $i^{\text{th}}$ column of W. Define the matrix $\overline{W}$ to be the matrix W that its $i^{\text{th}}$ column is multiplied by $a_i$ for $i = 1, \ldots, n$. Rewriting (9), we obtain

$$\mathcal{L}(\underline{a}, \overline{W}, \underline{b}) \qquad (9)$$

$$= \mathbb{E}_{\underline{x}, \underline{y}} \left\{ \left\| \underline{y} - \overline{W}\underline{x} - \underline{b} \right\|_2^2 \right\} + \sigma^2 \sum_{i=1}^n \overline{w}_i^2 \left( \frac{1}{a_i} - 1 \right)^2$$

where $\overline{w}_i$ is the $\ell_2$-norm of the $i^{\text{th}}$ column of $\overline{W}$. Using the Lagrange multiplier method for constrained optimization, we obtain

$$\frac{\partial}{\partial a_j} \mathcal{L}(\underline{a}, \overline{W}, \underline{b}) = -\lambda \frac{\partial}{\partial a_j} \|\underline{a}\|_\alpha^\alpha \qquad (10)$$

where $\lambda$ is a scalar and the right side of the equation is from the constraint in (7). Thus, we have

$$2\sigma^2 \overline{w}_j^2 \frac{1}{a_j^2} \left( \frac{1}{a_j} - 1 \right) = \lambda \, \alpha \, \text{sgn}(a_j) |a_j|^{\alpha-1} \qquad (11)$$

which leads to

$$\sigma^2 \overline{w}_j^2 \left( \frac{1}{a_j} - 1 \right)^2 = \frac{\lambda}{2} \alpha |a_j|^\alpha (1 - a_j). \qquad (12)$$

By summing over $j$'s and incorporating the constraint in (7), we get

$$\sigma^2 \sum_{j=1}^n \overline{w}_j^2 \left( \frac{1}{a_j} - 1 \right)^2 = \frac{\lambda}{2} \alpha \sum_{j=1}^n |a_j|^\alpha (1 - a_j)$$

$$= \frac{\lambda}{2} \alpha \left( k - \sum_{j=1}^n a_j |a_j|^\alpha \right) \qquad (13)$$

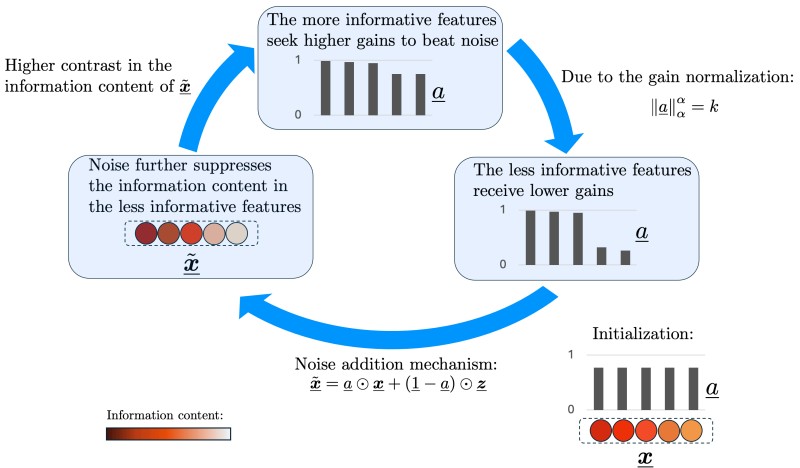

*Figure 2.* Reinforcing loop that results in polarization of the gains with $n = 5$ and $k = 3$.

Combining (13) and (10), we obtain

$$\mathcal{L}\left(\underline{a}, \overline{W}, \underline{b}\right) \tag{14}$$
$$= \mathbb{E}_{\underline{x},\underline{y}}\left\{\left\|\underline{y} - \overline{W}\underline{x} - \underline{b}\right\|_2^2\right\} + \frac{\lambda}{2}\alpha k - \frac{\lambda}{2}\alpha\sum_{i=1}^{n} a_i\left|a_i\right|^{\alpha}$$

Remember that we can confine the search space to the $a_i$'s between 0 and 1. Hence, according to (12), we have $\lambda \geq 0$. Therefore, the term at the end of (14) achieves its minima when there are $k$ of $a_i$'s equal to 1 and $n - k$ of them equal to 0, which completes the proof.

*Remark* 2.2. There are three hyper parameters in the SAND layer, $k$, $\sigma$ and $\alpha$:

- $k$ is the number of features to be selected. It will be initially set straightforwardly.

- $\sigma$ indicates how firmly we would like to restrict the number of features to $k$. A higher value of $sigma$ places greater emphasis on precisely achieving $k$ features, resulting in faster binarization (polarization toward 0 and 1) of the gains ($a_i$'s).

- $\alpha$ indicates which norm to be used to normalize the gain vector $\underline{a}$ during training.

We will see in the experiments that the method is not sensitive to the choice of $\sigma$ and $\alpha$. In fact, setting $\sigma$ within the range of standard deviation of the input features, and $\alpha = 2$, yields nearly optimal results across all datasets. Thus, there is no need to fine-tune $\sigma$ and $\alpha$. Moreover, to ensure stable training, we apply clipping to the $a_i$ values throughout the training process, keeping them within the range $[0, 1]$.

## 3. Experiments

**Feature Selection for Neural Networks**

We explored the performance of SAND through experiments on standard benchmark datasets used for feature selection in neural networks. Specifically, we utilized nine datasets, seven of which were used in previous studies by (Balın et al., 2019; Lemhadri et al., 2021; Yamada et al., 2020; Yasuda et al., 2023). The additional real and synthetic datasets were California Housing (Torgo, 1997) and HAR70 (Logacjov & Ustad, 2023). California (CA) Housing is a real dataset consisting of 20640 samples with 8 interpretable features, with the task of regressing the price of the house with the least features. Additionally, HAR70 is a large synthetic dataset comprising 2.3 million samples. It includes 6 informative features, which we augment with 100 nuisance features sampled from $\mathcal{N}(0, 0.1)$, and the task involves classifying the activity being performed. The testing metric for all datasets was accuracy, except for the CA Housing dataset, where the mean absolute error (MAE) was used. The datasets were normalized to have zero mean and unit standard deviation for each feature. Furthermore, we evaluated SAND on a novel multi-spectral image dataset, introduced in this paper for the first time, which has never been used before for feature selection. Details of this dataset and our evaluation are provided in Appendix C.

We implemented a neural network with one hidden layer and a ReLU activation. Given the variation in hidden layer widths across cited works, we opted for a width equal to $n/3$, where $n$ represents the dimensionality of the input data. Please refer to Table 3 of Appendix A for a comprehensive overview of the nine datasets, the corresponding number of

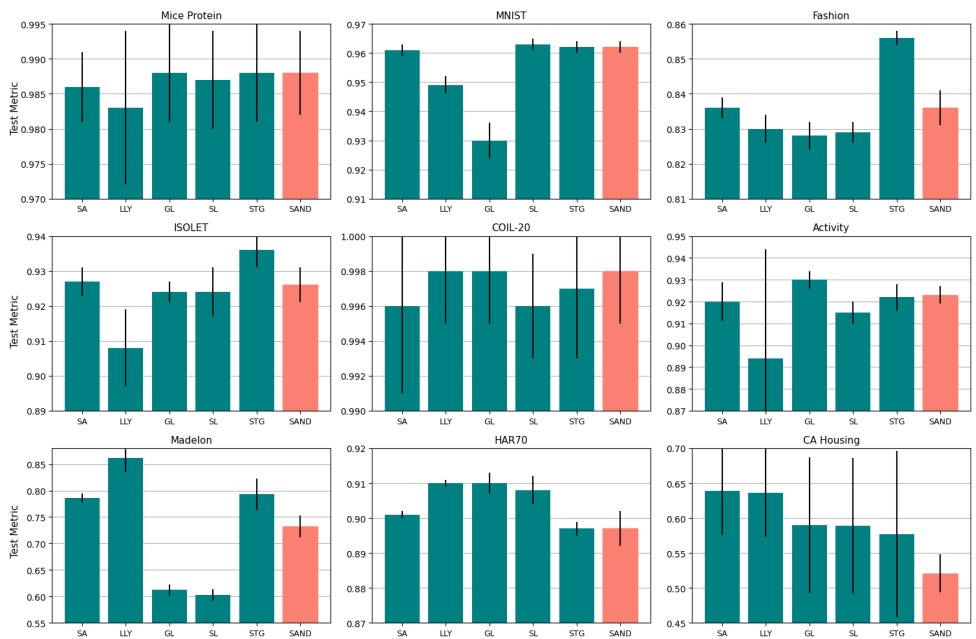

Figure 3. Test metrics on 9 datasets over 10 trials. The metric is accuracy (↑) for all except MAE (↓) for CA Housing being a regression problem. SA = Sequential Attention, LLY = Batch-wise Attenuation, GL = Group LASSO, SL = Sequential LASSO, and STG = Stochastic Gates.

epochs and batch size used for training, along with the mean accuracy/absolute-error of the model with all features.[2]

Our evaluation included a comparison between SAND and five established feature selection algorithms, namely Sequential Attention (SA) (Yasuda et al., 2023), Sequential LASSO (SL) (Luo & Chen, 2014), Btach-wise Attenuation (LLY) (Liao et al., 2021), Group LASSO (GL) (Zhao et al., 2015), and Stochastic Gates (STG) (Yamada et al., 2020).[3] For all methods except ours, training comprised a feature selection phase followed by a fitting phase wherein the neural network was retrained on the selected features. Although STG often eliminates the need for retraining thanks to its weight polarization, on some datasets (e.g. MICE Protein) the weights fail to polarize when $k = 60$, even after an exhaustive search over the regularization parameter which indirectly governs the number of selected features, and so we resort to retraining in some cases. As for SAND, the fitting phase was omitted and the weights learned during the selection phase were directly utilized for inference. In other words, the gains corresponding to the non-selected features where set to zero while keeping all other weights

of the model intact. Hence, from this point of view, our method offers two key benefits. Firstly, it demands fewer epochs (33% fewer epochs in our experiments). Secondly, it provides a streamlined pipeline where both selection and inference are handled by the same model.

Across all experiments, we employed the Adam optimizer with a learning rate of $10^{-3}$, and we partitioned the datasets into 70-10-20 splits for training, validation, and testing, respectively. For hyperparameters of the SAND layer, we used $\sigma = 1.5$ and $\alpha = 2$ consistently. Unless otherwise specified, we selected $k = 60$ features for all datasets by default, except for the following: $k = 5$ for the Madelon dataset, $k = 3$ for the CA Housing dataset, and $k = 6$ for the Har70 dataset. The comparative results are summarized in Figure 3, and the exact numerical values are shared in Table 4 of Appendix B. The error bars were calculated using the standard deviation over 10 trials. Drawing from the results in Figure 3, SAND competes effectively with other feature selection methods, while offering a streamlined pipeline that requires fewer iterations. It also exhibits a very consistent behavior as shown in Table 5 of Appendix B.

To provide additional insights, we varied the number of selected features $k$ under consistent settings and evaluated performance on the test set. Results are shown in Figure 4. Notably, SAND demonstrates its strength in feature selection, showcasing results that outperform or are comparable

---

[2]The code to reproduce our experiments is available at `https://github.com/csem/SAND`

[3]These algorithms represent the top-performing methods reported in the literature. A comparison with other approaches (Atashgahi et al., 2022; Lemhadri et al., 2021; Sokar et al., 2022) demonstrates that SAND achieves better performance.

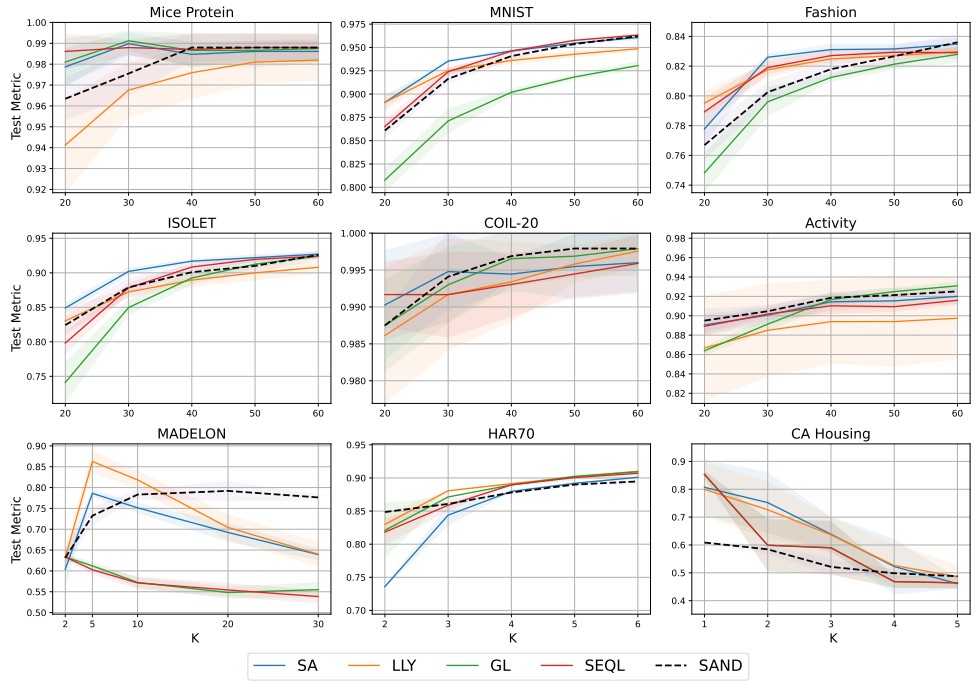

Figure 4. Test metrics for different $k$'s. Accuracy ($\uparrow$) for all except MAE ($\downarrow$) for CA Housing.

to other methods across different feature counts. This advantage is particularly significant given the fact that the best method is changing from dataset to dataset. Thus, there is a need for algorithms that deliver value beyond marginal accuracy improvements, prioritizing enhancements in computational demand and simplicity—qualities that our method exemplifies. It is important to note that all SAND experiments were conducted using fixed values of $\sigma = 1.5$ and $\alpha = 2$, without any fine-tuning. We omitted STG from Figure 4 due to the high computational cost of performing an exhaustive search over the regularization parameter $\lambda$ for each $k$. Moreover, on the MICE Protein dataset (77 features), STG fails to identify a clear feature subset once $k$ exceeds 37. For $k \leq 37$, STG produces a polarized solution—exactly $k$ features attain importance weight at almost 1, while the remainder are almost zero. However, for $k > 37$, the selection process stalls: irrespective of $\lambda$, only the same 37 features remain at weight 1, and the other 40 features converge to an almost uniform, nonzero weight. Adjusting $\lambda$ cannot induce the selection of additional features; it only uniformly increases the weights of these 40 unselected features above zero.

**Role of $\sigma$**

As discussed in Remark 2.2, the parameter $\sigma$ influences the rate of gain polarization. To demonstrate this, we trained SAND model on MICE dataset for 2000 epochs, using $\sigma$ values of 1.0 and 2.0, while keeping other settings fixed.

We recorded the gains every 10 epochs. Figure 5 presents the sorted gains for selected epochs. We observe that while the gains tend to cluster around 1 and 0 in both plots, this clustering occurs at a higher rate for a larger $\sigma$. Moreover, to assess SAND's sensitivity to $\sigma$, we replicated the initial experiment while varying $\sigma \in \{1.0, 1.5, 2.0, 2.5, 3.0\}$. Results are presented in Table 1. As evident in the table, our approach demonstrates high robustness to the selection of $\sigma$, which underscores a positive aspect of the proposed method.

**Effect of the choice of $\alpha$ in $\ell_\alpha$-Normalization**

To have an insight of the effect of $\alpha$, we conducted a duplicate experiment, this time employing $\alpha = 1.0$ (with $\sigma \in \{0.5, 1.5\}$). The outcomes are presented in Table 2. As shown in the table, the performance remains highly consistent, despite variations in $\alpha$.

## 4. Summary and Future Works

In this paper, we introduced a novel feature selection method. Specifically, we presented a new layer (SAND) that integrates into a neural network, enabling automatic feature selection during the training phase. The benefits of this approach include:

- On par with the state-of-the-art performance: Through extensive experiments, we showed that the proposed method has effectively state-of-the-art performance.

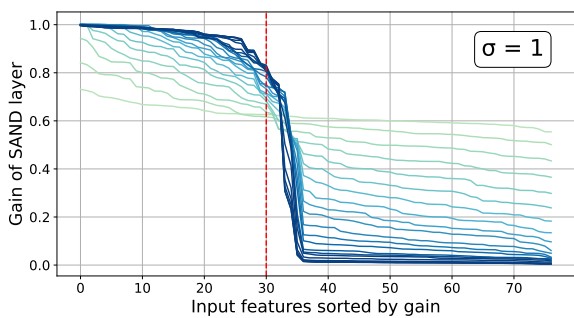 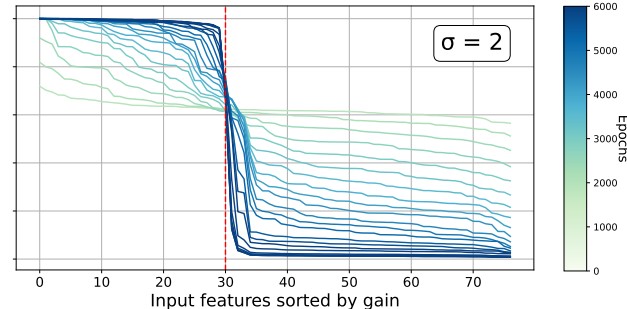

*Figure 5.* Polarization of the feature gains in SAND layer for $k = 30$.

*Table 1.* Test metrics for feature selection with SAND using different $\sigma$'s.

| Dataset | $\sigma = 1.0$ | $\sigma = 1.5$ | $\sigma = 2.0$ | $\sigma = 2.5$ | $\sigma = 3.0$ |
|---|---|---|---|---|---|
| Mice Protein | $0.988 \pm 0.006$ | $0.988 \pm 0.006$ | $0.988 \pm 0.007$ | $0.988 \pm 0.006$ | $0.987 \pm 0.006$ |
| MNIST | $0.953 \pm 0.007$ | $0.962 \pm 0.002$ | $0.958 \pm 0.002$ | $0.953 \pm 0.002$ | $0.948 \pm 0.004$ |
| MNIST-Fashion | $0.830 \pm 0.007$ | $0.832 \pm 0.007$ | $0.833 \pm 0.003$ | $0.831 \pm 0.004$ | $0.825 \pm 0.003$ |
| ISOLET | $0.913 \pm 0.009$ | $0.926 \pm 0.005$ | $0.922 \pm 0.007$ | $0.921 \pm 0.006$ | $0.916 \pm 0.006$ |
| COIL-20 | $0.991 \pm 0.005$ | $0.998 \pm 0.003$ | $0.998 \pm 0.002$ | $0.998 \pm 0.003$ | $0.996 \pm 0.004$ |
| Activity | $0.929 \pm 0.006$ | $0.923 \pm 0.004$ | $0.922 \pm 0.006$ | $0.924 \pm 0.004$ | $0.922 \pm 0.005$ |
| Madelon | $0.741 \pm 0.022$ | $0.732 \pm 0.021$ | $0.708 \pm 0.023$ | $0.689 \pm 0.017$ | $0.646 \pm 0.025$ |
| HAR70 | $0.901 \pm 0.002$ | $0.897 \pm 0.005$ | $0.889 \pm 0.002$ | $0.881 \pm 0.005$ | $0.873 \pm 0.006$ |
| CA Housing | $0.514 \pm 0.026$ | $0.521 \pm 0.027$ | $0.532 \pm 0.030$ | $0.557 \pm 0.022$ | $0.566 \pm 0.007$ |

- Low computational and memory burden: The layer introduces only $n$ trainable parameters, along with $n$ multiplication-additions and a single $n$-dimensional $\ell_2$-normalization, where $n$ is the number of features.

- One-shot feature selection and network training: There is no need for selecting the features in one phase of the training and then retrain the network with the selected features. Once the training phase has finished, the features are selected and the neural network is trained for the selected features.

- Control on the number of selected features: The number of features can be directly set in the algorithm in contrast to the main stream methods which require sweeping over a hyper parameter to be able to obtain the desired number of features.

- Considerably faster: As there is no need for the retraining phase, and due to the low computational overload, the method is considerably faster than the competitors.

- Handy Integration of Feature Selection in Neural Networks: Our feature selection method seamlessly integrates as an additional layer at the outset of the neural network, preserving the original architecture and loss function. With only input gradients required to train the layer gains, the network architecture or loss function can be treated as a black box.

- Tailored features to the application and the neural network architecture: Since SAND layer is an integral component of the base model, the features selected are automatically adapted for the specific application at hand and the chosen model architecture.

- Remarkably simple both conceptually and practically: the mathematical model of our method involves only entrywise multiplication, addition with Gaussian noise, followed by $\ell_2$-norm normalization, rendering it remarkably simple in theory and in practice.

It is worth mentioning that the proposed SAND layer works in a very similar way to the Dropout layer but with an opposing effect. In the Dropout layer, randomization leads to an even distribution of information across all neurons. Conversely, randomization in SAND, due to weights' constraint, selects only neurons with the highest information content.

A straightforward continuation of this work is to explore

SAND layer's performance for network pruning by incorporating it into intermediate layers, akin to how Dropout and Batch Normalization layers are utilized. Additionally, studying the effect of different noise distributions and rigorous understanding of the effect of $\alpha$ and $\sigma$ for different network architectures (dense, convolutional, transformers, etc.) are interesting lines for future researches. Furthermore, considering features relation structure during the selection is another valuable avenue to explore.

Table 2. Effect of $\alpha$ on SAND layer.

| Dataset | $\alpha = 1.0$ | | $\alpha = 2.0$ |
|---|---|---|---|
| | $\sigma = 0.5$ | $\sigma = 1.5$ | |
| Mice Protein | $0.987 \pm 0.005$ | $0.988 \pm 0.007$ | $0.988 \pm 0.006$ |
| MNIST | $0.959 \pm 0.002$ | $0.956 \pm 0.002$ | $0.962 \pm 0.002$ |
| MNIST-Fashion | $0.828 \pm 0.007$ | $0.830 \pm 0.006$ | $0.832 \pm 0.007$ |
| ISOLET | $0.922 \pm 0.008$ | $0.910 \pm 0.010$ | $0.926 \pm 0.005$ |
| COIL-20 | $0.997 \pm 0.003$ | $0.996 \pm 0.005$ | $0.998 \pm 0.003$ |
| Activity | $0.922 \pm 0.004$ | $0.907 \pm 0.008$ | $0.923 \pm 0.004$ |
| Madelon | $0.756 \pm 0.059$ | $0.675 \pm 0.029$ | $0.732 \pm 0.021$ |
| HAR70 | $0.854 \pm 0.036$ | $0.814 \pm 0.038$ | $0.897 \pm 0.005$ |
| CA Housing | $0.538 \pm 0.063$ | $0.560 \pm 0.023$ | $0.521 \pm 0.027$ |

## Acknowledgements

The present work was developed within the AGRARSENSE Project that has received Chips JU funding (Grant Agreement No. 101095835). It has received funding from the Swiss State Secretariat for Education, Research and Innovation (SERI) and is co-funded by the Innosuisse – Swiss Innovation Agency. More information: info@agrarsense.eu.

## Impact Statement

This paper presents work whose goal is to advance the field of Machine Learning. There are many potential societal consequences of our work, none which we feel must be specifically highlighted here.

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

## A. Experimental set-up

We provide Table 3 containing details about all datasets utilized in the feature selection experiments. Additionally, the table includes the epochs employed during training for each dataset, identifying the ones used for feature selection and the ones used to retrain/fit the model on the selected features. As indicated in the experiments (Section 3), fitting epochs are only utilized by models other than SAND, whereas SAND employs only the 'Select Epochs'. The table also presents the test accuracy of the base model trained using all features.

*Table 3.* Dataset characteristics, experiment parameters, and all-features performance metrics.[5]

| Dataset | (n, d) | # Classes | Select Epochs | Fit Epochs | Batch Size | All Features |
|---|---|---|---|---|---|---|
| Mice Protein | (1,080, 77) | 8 | 400 | 200 | 64 | $0.987 \pm 0.006$ |
| MNIST | (70,000, 784) | 10 | 100 | 50 | 64 | $0.978 \pm 0.001$ |
| MNIST-Fashion | (70,000, 784) | 10 | 200 | 100 | 64 | $0.878 \pm 0.003$ |
| ISOLET | (7,797, 617) | 26 | 400 | 200 | 64 | $0.958 \pm 0.002$ |
| COIL-20 | (1,440, 400) | 20 | 1000 | 500 | 64 | $0.996 \pm 0.003$ |
| Activity | (10,299, 561) | 6 | 200 | 100 | 64 | $0.941 \pm 0.002$ |
| Madelon | (2,600, 500) | 2 | 500 | 250 | 64 | $0.575 \pm 0.017$ |
| HAR70 | (2,259,597, 106) | 8 | 6 | 3 | 64 | $0.890 \pm 0.002$ |
| CA Housing | (20,640, 8) | N/A | 200 | 100 | 64 | $0.440 \pm 0.011$ |

Moreover, the experiments were executed on a machine equipped with an NVIDIA GeForce RTX 4090 GPU with 24GB of RAM, paired with an AMD Ryzen 9 5900X 12-Core Processor featuring 24 threads. The code to reproduce our experiments is available at `https://github.com/csem/SAND`.

## B. Experimental results

We present in Table 4 the benchmarking results of SAND alongside other methods on the nine datasets discussed in the experiments (Section 3). The intervals were calculated using the standard deviation across 10 trials.

*Table 4.* Test metrics over 10 trials (mean $\pm$ standard deviation): Accuracy ($\uparrow$) for all except MAE ($\downarrow$) for CA Housing.

| Dataset | SA | LLY | GL | SL | STG | SAND |
|---|---|---|---|---|---|---|
| Mice Protein | $0.986 \pm 0.005$ | $0.983 \pm 0.011$ | $0.988 \pm 0.007$ | $0.987 \pm 0.007$ | $0.988 \pm 0.007$ | $\mathbf{0.988} \pm 0.006$ |
| MNIST | $0.961 \pm 0.002$ | $0.949 \pm 0.003$ | $0.930 \pm 0.006$ | $\mathbf{0.963} \pm 0.002$ | $0.962 \pm 0.002$ | $0.962 \pm 0.002$ |
| MNIST-Fashion | $0.836 \pm 0.003$ | $0.830 \pm 0.004$ | $0.828 \pm 0.004$ | $0.829 \pm 0.003$ | $\mathbf{0.856} \pm 0.002$ | $0.836 \pm 0.005$ |
| ISOLET | $0.927 \pm 0.004$ | $0.908 \pm 0.011$ | $0.924 \pm 0.003$ | $0.924 \pm 0.007$ | $\mathbf{0.936} \pm 0.005$ | $0.926 \pm 0.005$ |
| COIL-20 | $0.996 \pm 0.005$ | $0.998 \pm 0.003$ | $0.998 \pm 0.003$ | $0.996 \pm 0.003$ | $0.997 \pm 0.004$ | $\mathbf{0.998} \pm 0.003$ |
| Activity | $0.920 \pm 0.009$ | $0.894 \pm 0.050$ | $\mathbf{0.930} \pm 0.004$ | $0.915 \pm 0.005$ | $0.922 \pm 0.006$ | $0.923 \pm 0.004$ |
| Madelon | $0.786 \pm 0.008$ | $\mathbf{0.862} \pm 0.027$ | $0.612 \pm 0.010$ | $0.603 \pm 0.011$ | $0.793 \pm 0.03$ | $0.732 \pm 0.021$ |
| HAR70 | $0.901 \pm 0.001$ | $\mathbf{0.910} \pm 0.001$ | $0.910 \pm 0.003$ | $0.908 \pm 0.004$ | $0.897 \pm 0.002$ | $0.897 \pm 0.005$ |
| CA Housing | $0.639 \pm 0.063$ | $0.636 \pm 0.063$ | $0.590 \pm 0.097$ | $0.589 \pm 0.097$ | $0.577 \pm 0.119$ | $\mathbf{0.521} \pm 0.027$ |

Table 5 presents the consistency analysis for the HAR70 dataset, which comprises 6 informative features augmented with 100 synthetically generated noisy ones. This dataset was specifically chosen because SAND demonstrates its lowest relative performance here compared to other methods, as shown in Table 4 (with SAND's classification accuracy being $0.013\%$ lower than the best-performing method, LLY). Nevertheless, SAND exhibits high consistency, consistently selecting 5 out of the 6 informative features and misidentifying the remaining feature only $20\%$ of the time. Compared to SA and SL, SAND is more reliable in selecting the informative features. However, its slightly lower performance can be attributed to the inherent randomness of the experiments and the fact that other methods benefit from retraining after feature selection, unlike SAND, which operates without this additional training phase.

---

[5]The metric is classification accuracy ($\uparrow$) for all except MAE ($\downarrow$) for CA Housing. Our study utilizes the entire MNIST and Fashion datasets, unlike related works. Additionally, the Activity dataset sourced from (Lemhadri et al., 2021)'s Google Drive and (Yasuda et al., 2023)'s repository contains 10,299 samples, as opposed to the 5,744 samples reported in the referenced papers.

*Table 5.* Selection consistency on HAR70: feature selection frequency of the useful features $(0 - 5)$ and the misselected ones over 10 runs for $k = 6$.

| Feature | SA | SL | **SAND** | STG | GL | LLY |
|---|---|---|---|---|---|---|
| 0 | 10 | 10 | 10 | 10 | 10 | 10 |
| 1 | 8 | 9 | 8 | 8 | 9 | 10 |
| 2 | 10 | 10 | 10 | 10 | 10 | 10 |
| 3 | 10 | 10 | 10 | 10 | 10 | 10 |
| 4 | 10 | 7 | 10 | 10 | 10 | 10 |
| 5 | 2 | 10 | 10 | 10 | 10 | 10 |
| Misselected | 10 | 4 | 2 | 2 | 1 | 0 |

## C. Benchmarking on real-world multi-spectral image dataset

In this section, we introduce a novel real-world multi-spectral image dataset, where feature selection is crucial for developing efficient and cost-effective acquisition hardware. We apply SAND to this dataset and demonstrate that it outperforms state-of-the-art methods.

Multi-spectral imaging is a powerful technology that captures images across multiple wavelengths, unlike conventional RGB imaging, which may fail to reveal critical spectral information. This approach has shown significant promise in various applications, including cancer detection in medical imaging (Ortega et al., 2020), precision agriculture (ElMasry et al., 2019), and chemometrics (Dupont et al., 2020). There are multiple ways to acquire multi-spectral data, including snapshot cameras, push-broom scanners, and filter wheel systems (Spigulis, 2024). However, a more cost-effective and flexible approach involves actively controlling the illumination. This method uses a monochromatic camera while sequentially activating lights at different wavelengths, enabling multi-spectral acquisition without requiring specialized optical components (Dunbar et al., 2020).

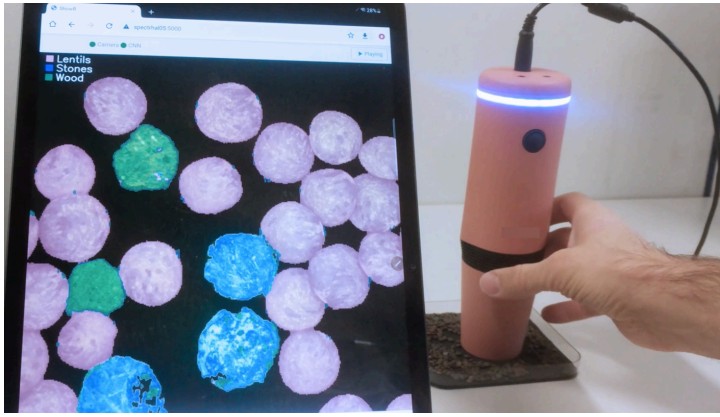

*Figure 6.* Demo of real-time segmentation on the MSI Grain dataset using multi-spectral imaging.

The "MSI Grain" dataset was acquired using active LED light control across 15 narrow spectral bands ranging from 360 nm to 940 nm. It contains spectral data for four distinct classes: Lentils, Wood, Stones, and Background. The goal is to leverage multi-spectral imaging to classify these visually similar classes, which are difficult to distinguish with the naked eye or conventional imaging. The dataset consists of 50,000 spectra per class, extracted from several multi-spectral image cubes (see Figure 6). Each spectrum corresponds to a 15-dimensional pixel within the image, representing its spectral signature

across the measured wavelengths. The dataset intentionally includes only spectral information, excluding spatial context. This design choice ensures that classification relies solely on the material's spectral response rather than its shape. The dataset is available in the code repository at https://github.com/csem/SAND.

A key challenge in multi-spectral imaging is reducing the number of spectral bands, or features, to optimize hardware efficiency. Fewer bands help minimize heat dissipation, lower hardware costs for commercial adoption, and enable fast real-time image acquisition. As a result, feature selection is not just a theoretical challenge but a practical necessity in this domain. SAND was developed to address this real-world need by jointly optimizing machine learning-based spectral detection and hardware design. Conventional feature selection methods often fall short in terms of performance and consistency, while state-of-the-art approaches tend to be complex and less user-friendly. In contrast, SAND provides a practical, application-driven solution inspired by these challenges, reinforcing its effectiveness in real-world scenarios.

We present in Table 6 the benchmarking results of SAND alongside other methods on the MSI Grain dataset. The target number of selected features is $k = 9$ and the experiments were run across 5 trials. All other experimental set-ups and pre-processing steps are the same as discussed in Section 3. Along with its remarkable simplicity compared to other methods, SAND achieves the highest performance, as demonstrated below.

*Table 6.* Test accuracy over 5 trials on MSI Grain (mean $\pm$ standard deviation).

| Method | SA | LLY | GL | SL | STG | SAND |
|---|---|---|---|---|---|---|
| Accuracy | $0.917 \pm 0.003$ | $0.917 \pm 0.001$ | $0.916 \pm 0.002$ | $0.918 \pm 0.002$ | $0.911 \pm 0.005$ | $\mathbf{0.919} \pm 0.002$ |
| All Features Accuracy | $0.924 \pm 0.001$ | | | | | |

It is important to note that the segmentation in Figure 6 occurs in real-time, with the trained neural network running directly on the handheld edge device capturing the image. No cloud computing is involved—the tablet is used solely for visualization. Real-time edge processing was made possible by feature selection, which reduced the number of relevant spectral bands to nine, significantly decreasing both the model size and the number of spectral acquisitions. This optimization enabled the algorithm to be deployed on the edge device and operate in real-time.

