# OpenReview forum: "SAND: One-Shot Feature Selection with Additive Noise Distortion"
_ICML.cc/2025/Conference — ICML 2025 poster_

### Official Review · Reviewer_x6bu · 2025-03-13

**Overall Recommendation:** 2

**Summary:**

This paper proposes SAND, a new feature selection method which modifies the original input to a linear combination of additive zero-mean Gaussian noises and the original input. The weight of linear combination $a\in \mathbb{R}^d$ measures the feature importance and the constraint of the number of selected features is imposed in a soft way by requiring the $\alpha$-norm of $a$ equaling $k$. By doing so, there is no need to add regularization term to the original loss function, which can be optimized using popular gradient descent methods like Adam. In the inference stage, only features within the top-$k$ weights $a$ are retained and used as the inputs to the trained model. Experiments on synthetic and real datasets demonstrate the effectiveness of the proposed method compared to existing methods including sequential attention, batch-wise attention, sequential Lasso and group Lasso.

**Claims And Evidence:**

The claim that SAND achieves SOTA performance on benchmark datasets is not well supported. SAND performs worse than other methods especially in the case with small $k$ in experiments.

**Essential References Not Discussed:**

To the best of my understanding, key references are reviewed in this paper.

**Experimental Designs Or Analyses:**

I have check the soundness of experimental designs.

**Methods And Evaluation Criteria:**

The proposed method is well motivated and the evaluation criteria makes sense. However, SAND is very similar to STG [34] since both of them introduces randomness in training, but the comparison to STG is missed in experiments.

**Other Comments Or Suggestions:**

None

**Other Strengths And Weaknesses:**

Strengths:
1. The proposed method is well motivated.

Weakness:
1. On the benchmark datasets, the performance of SAND is not the SOTA compared to other methods, especially when $k$ is small.
2. Writing can be further improved, especially the analysis of SAND in the linear regression case.

**Questions For Authors:**

My concerns are listed in the weakness section.

**Relation To Broader Scientific Literature:**

Apart from feature selection, the idea of introducing noise to the input variables has been investigated in prior feature noising method, which introduces either multiplicative or additive noise to the input variables to enhance the generalization ability and robustness of the model.

[1] Maaten L, Chen M, Tyree S, et al. Learning with marginalized corrupted features[C]//International Conference on Machine Learning. PMLR, 2013: 410-418.
[2] Wager S, Wang S, Liang P S. Dropout training as adaptive regularization[J]. Advances in neural information processing systems, 2013, 26.
[3] Zhuo J, Zhu J, Zhang B. Adaptive Dropout Rates for Learning with Corrupted Features[C]//IJCAI. 2015, 24: 4126-4133.
[4] Li Z, Gong B, Yang T. Improved dropout for shallow and deep learning[J]. Advances in neural information processing systems, 2016, 29.
[5]

**Theoretical Claims:**

I have checked the analysis of SAND in the linear regression case. The last term in Eq. (14) achieves its minima when there are k of $a_i$'s equal to 1 is not obvious and detailed analysis is needed.

---

> ### Author Rebuttal · Authors · 2025-03-30
>
> Thank you for the careful reading of the paper and your valuable feedback.
>
> Regarding being SOTA, our work emphasizes that although no single method consistently outperforms all others across every dataset, our approach is notably simpler and competitive—especially when accuracy/MAE metrics are saturated (see Table 4 in Appendix B). We believe that a method that is the best in a significant number of cases and nearly optimal in the remaining ones merits the designation of state-of-the-art—an acknowledgment that can apply to multiple methods when performance differences are marginal (SAND performs better or is comparable with other methods although it is extremely simpler conceptually and considerably faster in practice: no tuning/line-search to get to k features, no alteration of loss function, no post-selectin training). Nonetheless, we recognize that the wording may have been misleading, and we will revise it in the final version of the paper to better reflect these nuances.
>
> For small k, no state-of-the-art method consistently outperforms the rest. Specifically, SAND excels on three datasets for small k’s; for example, on the CA Housing dataset (regression, lower MAE is better), SAND shows superior performance for small k. Moreover, SAND is trained with about 33% fewer epochs than competitors, as it requires no post-selection training. This efficiency may explain its underperformance in some cases, as detailed in Appendix B (e.g., the analysis on Har70, chosen for its known correct features and relatively lower SAND performance). Although SAND’s classification accuracy on Har70 is 0.013% lower than the best (LLY), it achieves better feature selection accuracy (correct features out of 6) and greater stability than the two methods (SA and SL) that slightly outperform it in classification
>
> Thank you for the suggested references. We can add them to the final version of the paper.
>
> Concerning the proof in linear regression, since $a_i$ s are between 0 and 1,  for each $i$, $a_i |a_i|^\alpha <= |a_i|^\alpha$ with equality holding only for $a_i=0$ or $a_i=1$. Thus, we have $\sum_i{ a_i |a_i|^\alpha } <= \sum_i{|a_i|^\alpha } = k$ with euqlity holds only if all $a_i$ s are either 0 or 1. Since in the first two terms of Eq. (14), there’s no a_i, and \lambda is positive, the whole expression will be minimized when $\sum_i{ a_i |a_i|^\alpha }$ gets its maximum value (k) and this happens when all of $a_i$ s are either 0 or 1 . Since we have the constraint $\sum_i{|a_i|^\alpha } = k$, we conclude that exactly k of $a_i$ s are 1 and (n-k) of $a_i$ s are 0 . We can add this explanation to the final version of the paper and improve the writing, while respecting the page limit.
>
> Concerning the comparison with STG: we tried to compare our method with other state-of-the-art methods, some of which were proposed very recently. Furthermore, a quick comparison was already done with other methods, including STG and LassoNet, as mentioned in footnote 3 (page 6, lines 326-328). However, this comparison was not quantitatively elaborated in the current version of the paper. We agree that a direct quantitative comparison with STG method is useful, given the similarity with SAND (SAND is much simpler with no need for loss function alteration or hyperparameter search for k as in SGT). For this end, we have now included the STG in our previously shared anonymized code repo (https://anonymous.4open.science/r/SAND-6BB1), which allows us to compare it with SAND and the 4 other methods used in our paper for benchmarking. Furthermore, the readme file of the repo now has a table comparing SAND and STG on 5 classification datasets and one regression dataset. SAND outperforms STG on 3 classification datasets and on the regression dataset (lower MAE is better). Note that for the 2 datasets on which SAND is worse, SAND was already not presented as the top performer (in terms of accuracy) in the current version of the paper.
> These 6 datasets were chosen for their moderate size (less than 70k samples) and for fast experimentation in this rebuttal. This is because STG needs a computationally costly fine-grained tuning to select the specified number of parameters. In the final version of the paper, we can include the comparison with STG on all 9 datasets.
>
> We kindly ask the reviewer to consider these clarifications and to reach out during the discussion phase if further explanation is needed. We reiterate the potential of this simple, practical, and efficient method, which may impact areas beyond feature selection (e.g., NN pruning). Our goal is not to introduce a model that dramatically outperforms all baselines but to present a lightweight model that at least matches baseline performance while offering distinct advantages: simplicity, low computational and memory burden, one-shot feature selection and network training, control over the number of selected features, and easy integration with neural networks.

---

### Official Review · Reviewer_UP58 · 2025-03-15

**Overall Recommendation:** 3

**Summary:**

The paper introduces SAND (Selection with Additive Noise Distortion), a novel feature selection layer for neural networks that automatically selects $k$ informative features during training. SAND operates by multiplying each input feature by a trainable gain $a_i$ while adding Gaussian noise weighted by $1-a_i$. Such a design encourages the informative $k$ gains to 1 (selecting features) and the rest to 0 (discarding features) without altering the loss function or network architecture. Unlike traditional methods, SAND requires no post-selection retraining or extensive hyperparameter tuning. Experimental results on nine benchmark datasets (e.g., MNIST, ISOLET) as well as a new real-world multi-spectral imaging dataset (MSI Grain) demonstrate that SAND matches or exceeds state-of-the-art methods in accuracy and efficiency. A theoretical analysis in linear regression further validates its approach, positioning SAND as a simple yet powerful tool for feature selection.

**Claims And Evidence:**

This paper has several claims. While majority of them are well-supported. The claim that *SAND directly controls the number of selected features without hyperparameter tuning* is bit overclaimed. SAND requires to explicitly define the number of selected features $k$, which is usually missing in real-world dataset. Other approaches from feature interpretation domains can only explicitly define the number $k$ to conduct feature selection.

**Essential References Not Discussed:**

Not that I know of.

**Experimental Designs Or Analyses:**

The dataset selection is a bit questionable. The only large-scale dataset is Har70, which has a number of well-known informative features. Based on the hyper-parameter selection in Section 3, such information helps authors better determine $k$. Evaluating SAND on large-scale datasets without a number of informative features pre-known, such as Criteo or ImageNet (could be compressed), can better showcase SAND's ability in real-world scenarios. The paper also misses baselines from discrete optimization domains, which can naturally be adapted for feature selection.

**Methods And Evaluation Criteria:**

Both the methods and evaluation criteria are generally reasonable.

**Other Comments Or Suggestions:**

No

**Other Strengths And Weaknesses:**

Pros:
1. This work tackles an interesting and applicable problem.
2. The writing of this paper is generally easy to follow.

Cons:
1. The experimental setup could be questionable, which limits its contribution and applicability.
2. Certain claims are a bit over-claimed and not unique.

**Questions For Authors:**

No

**Relation To Broader Scientific Literature:**

One-shot feature selection without post-selection retraining is a very interesting topic that can be applicable to the real world. The reviewer finds this point being the most important contribution of this paper.

**Theoretical Claims:**

I quickly check its correctness, but I am not an expert in theory.

---

> ### Author Rebuttal · Authors · 2025-03-30
>
> Thank you for the careful reading of the paper and the valuable comments.
>
> We agree that finding out the required number of features for a desired performance is a very valuable question to answer, but it is outside the scope of this work. By the sentence “SAND directly controls the number of selected features without hyperparameter tuning”, we mean that once the number of desired features k is given, the algorithm directly gives a solution for that many features. This is in contrast with other approaches where, even when k is given, there is another hyper parameter by which we can push the algorithm to have more or less number of features, and there is no direct control on the number of features to select in one shot. In LASSO-based approaches, stochastic gates, and many other methods, there is no direct control over the number of features. There, we can only achieve the exact (predefined) number of desired features by sweeping over the other hyper parameter (which is usually a regularization term in the loss). In short, for a given k (whether defined by the hardware constraints as explained in the dataset of Appendix C, or by model complexity/size constraint,...), SAND finds/selects the k features in one shot without a sweeping or line search, in contrast with many other methods. The fact that k is a predefined number in the context considered here is mentioned in the abstract, but this might get confusing later on in the paper. We will make sure to clarify this confusion in the final version of the paper by stressing that this is the standard feature selection problem where k is given and we are after selecting the k features (not identifying k itself).
>
> We acknowledge that expanding the range of datasets and experimental scenarios can further strengthen the evaluation of a newly proposed method. In our work, we deliberately selected datasets that are widely recognized as standard benchmarks for feature selection, ensuring a fair and consistent basis for comparison within a reasonable computational timeframe. Moreover, we compared our approach against state-of-the-art methods, including those acclaimed in the literature, such as the recent work by Google on sequential attention feature selection. In addition to these benchmarks, we introduced a novel real-world dataset (detailed in Appendix C) to demonstrate the practical applicability of our method. We believe that the breadth of datasets and competitive methods included in our study represents one of the most comprehensive evaluations in the current literature. This selection was carefully designed to balance thoroughness with computational feasibility while ensuring a robust comparison with leading approaches.
>
> We appreciate your feedback and hope that the clarifications provided have been helpful. Should you need any further details during the rebuttal phase, please feel free to reach out, and we kindly invite you to reconsider our work in light of these explanations.

---

> > ### Comment · Reviewer_UP58 · 2025-04-03
> >
> > Thanks for the feedback. I will maintain my score. Good luck

---

### Official Review · Reviewer_PP3t · 2025-03-22

**Overall Recommendation:** 4

**Summary:**

This paper proposes SAND (Stochastic Additive Noise Decoupling), a method for feature selection that avoids adding any explicit loss or regularization term. Instead, it uses a simple noise-injection mechanism where each input feature is blended with noise according to a learnable gating parameter $a$. The gating vector is constrained via an $\ell_\alpha$-norm, and features with higher $a_i$ values are preserved with less noise. The method is evaluated across several datasets and shows performance that is comparable to existing feature selection approaches.

**Claims And Evidence:**

Most claims are supported by experiments and theoretical reasoning. However, an important gap is the lack of discussion around the training behavior of $a_i$ values. While the paper shows that optimal solutions lie in $[0,1]$, there are no guarantees or constraints during training that enforce this. Since the model is trained with gradient-based optimization, it's unclear how $a_i$ values are kept within $[0,1]$ in practice. This should be clarified, especially since negative or $>1$ values could affect the noise interpolation behavior.

**Essential References Not Discussed:**

No.

**Experimental Designs Or Analyses:**

The experiments are straightforward and use commonly adopted architectures and datasets.

**Methods And Evaluation Criteria:**

Yes, the methods and evaluation criterion are appropriate. The datasets and architectures used are standard, and the comparisons to existing feature selection methods are relevant.

**Other Comments Or Suggestions:**

N/A

**Other Strengths And Weaknesses:**

Strengths:
 1. The approach is simple and can be easily integrated into standard training pipelines.
 2. Avoids the need for tuning additional loss weights or regularization terms.

Weaknesses:
1. It is unclear how the model ensures that $a_i \in [0,1]$ during training. The paper relies on theoretical reasoning that such a solution exists, but does not explain how it's maintained during optimization. Since gradient updates can move parameters outside this range, it’s important to address this empirically or with a constraint mechanism.
2. Empirical comparison to stochastic gates would make the evaluation more complete, as they also perform embedded feature selection using a gating mechanism, but through loss regularization. This would help clarify how SAND compares in practice and in formulation.

**Questions For Authors:**

1. Are the $a_i$ values explicitly constrained or reparameterized during training to ensure they stay within $[0,1]$? If not, how is it ensured that values do not become negative or exceed 1 during optimization?
2. How does SAND compare to stochastic gates in practice, especially since both aim to select features using differentiable mechanisms? Including this comparison would help contextualize the method.
3. An analysis of the stability or sparsity of the learned $a$ vectors across different runs or initializations could inform the robustness of the approach. Including such an analysis would enhance the completeness of the paper.

**Relation To Broader Scientific Literature:**

The work is related to prior methods in differentiable feature selection, such as stochastic gates and other gating mechanisms that involve regularization. The key difference is that SAND avoids adding terms to the loss and instead relies on noise injection and norm constraints, making it a simpler alternative.

**Theoretical Claims:**

I have verified the derivation provided for the linear regression case, where the authors show that adding the SAND layer is equivalent to introducing a term in the loss function that promotes the selection of $k$ features. The steps and conclusions appear to be correct and are consistent with the behavior expected from the formulation.

---

> ### Author Rebuttal · Authors · 2025-03-30
>
> Thank you for the careful reading of the paper and the valuable comments.
>
> Concerning the first question, as mentioned by the reviewer, mathematically, the optimum a_i is guaranteed to be between 0 and 1. In the code, to be on the safe side, we always clip the values between 0 and 1 at each iteration when the vector a_i is normalized. Now that the issue is cleverly raised as a concern by the reviewer, we will add it to the text of the paper as well.
>
> Concerning the second question, we tried to compare our method with other state-of-the-art methods, some of which were proposed very recently. Furthermore, a quick comparison was already done with other methods, including stochastic gate and LassoNet, as mentioned in footnote 3 (page 6, lines 326-328). However, this comparison was not quantitatively elaborated in the current version of the paper. We agree that a direct quantitative comparison with the stochastic gates method might be useful, given the similarity with SAND (note that SAND is much simpler with no need for loss function alteration or hyperparameter search for k as in stochastic gates ). For this end, we have now included the stochastic gates (STG) in our previously shared anonymized code repo (https://anonymous.4open.science/r/SAND-6BB1), which allows us to compare it with SAND and the 4 other methods used in our paper for benchmarking. Furthermore, the readme file of the repo now has a table comparing SAND and STG on 5 classification datasets and one regression dataset. SAND outperforms STG on 3 classification datasets and on the regression dataset (lower MAE is better). Note that for the 2 datasets on which SAND is worse, SAND was already not presented as the top performer (in terms of accuracy) in the current version of the paper.
> These 6 datasets were chosen for their moderate size (less than 70k samples) and for fast experimentation in this rebuttal. This is because STG needs a computationally costly fine-grained tuning to select the specified number of parameters. In the final version of the paper, we can include the comparison with STG on all 9 datasets.
>
>
> For the third question, a quick robustness/consistency experiment was done in Table 5 of Appendix B (supplementary material). The experiment was purposely done on one dataset with known correct features and on which SAND performs the worst compared to other methods (as explained in the appendix). Nevertheless, SAND shows a high consistency in choosing the correct features: SAND has better consistency and more accurate feature selection than the two methods that seem to outperform it in terms of classification accuracy (note that SAND is trained with 33% less number of epochs compared to other methods since it does not require post-selection training; that might be the reason it appears to undeperform in some scenarios as explained in the appendix). Such robustness analysis can indeed be extended to other datasets (even if the correct features are not known a priori) to track and confirm the robustness/consistency of SAND. We can run the experiment more thoroughly and add it to the supplementary material (or the paper, should we have enough space).
>
> Thanks for the productive suggestions. We hope these clarifications prove useful, and please feel free to reach out during the rebuttal phase if further discussion is needed; we kindly ask that you reconsider your evaluation in light of these explanations.

---

> > ### Comment · Reviewer_PP3t · 2025-04-08
> >
> > Thank you for your response. Your answers addressed my questions, so I will be update my score to accept.

---

### Official Review · Reviewer_gNnB · 2025-03-22

**Overall Recommendation:** 4

**Summary:**

The key contribution of this paper is to introduce a method for feature selection during training of a neural network. A key feature of this method is to be able to do the feature selection in a very simple way, without substantive modifications to the architecture itself, while still being able to retain good performance. The main idea is to have a trainable noise that can be added to the feature, which forces the features to be clustered at the end of the training process.

**Claims And Evidence:**

I would say so, it feels like the paper is well-written and thought through. The examples chosen are perhaps a bit small-scale but still provide an illustration of the efficacy of the method.

**Essential References Not Discussed:**

Not to the best of my knowledge.

**Experimental Designs Or Analyses:**

I did. I don't think with the experiments as presented there are any issues.

**Methods And Evaluation Criteria:**

The methods and the selected data sets used in the evaluation are appropriate.

**Other Comments Or Suggestions:**

N/A

**Other Strengths And Weaknesses:**

N/A

**Questions For Authors:**

Is it possible to make k a learnable parameter, somehow?

**Relation To Broader Scientific Literature:**

I think in the crowded and extensively studied problem of feature selection, the authors managed, to the best of my knowledge, make a nice contribution to the literature.

**Theoretical Claims:**

I did. The authors presented a nice illustration with linear regression showing how their proposed method effectively promotes the selection of six features.

---

> ### Author Rebuttal · Authors · 2025-03-30
>
> We appreciate your careful reading of our paper and your kind compliments. Regarding your question about learnable k, for the moment what comes to our mind is the simple idea to sweep over k and see where the loss exhibits a drastic change.

---

### Decision · Program_Chairs · 2025-05-01

**Decision:**

Accept (poster)

**Comment:**

This paper proposed stochastic additive noise decoupling (SAND) for feature selection which can avoid adding explicit loss or regularization term via simple noise-injection mechanism. Reviewers praised that the proposed approach is simple, can be easily integrated into standard training pipelines and avoid tuning additional regularization parameter or feature selection parameter. Reviewers also raised some concerns such as the comparison to stochastic gates, the sensitivity of parameter a_i, and the selection of datasets. Authors’ responses to these concerns are satisfactory which addressed critical issues like competitive results against stochastic gate on 6 datasets and the reported sensitivity analysis of a_i.. As a novel alternative to stochastic gate approach, the acceptance is recommended.